# Laser Therapy in Heavily Treated Oncological Patients Improves Vaginal Health Parameters

**DOI:** 10.3390/cancers16152722

**Published:** 2024-07-31

**Authors:** Marco Di Stanislao, Camelia Alexandra Coada, Francesca De Terlizzi, Stella Di Costanzo, Enrico Fiuzzi, Francesco Mezzapesa, Giulia Dondi, Damiano Paoli, Gianluca Raffaello Damiani, Francesco Raspagliesi, Giorgio Bogani, Antonino Ditto, Alessio Giuseppe Morganti, Pierandrea De Iaco, Anna Myriam Perrone

**Affiliations:** 1Division of Oncologic Gynecology, IRCCS Azienda Ospedaliero-Universitaria di Bologna, 40138 Bologna, Italy; marco.distanislao@studio.unibo.it (M.D.S.); enrico.fiuzzi@studio.unibo.it (E.F.); francesco.mezzapesa@studio.unibo.it (F.M.); giulia.dondi@aosp.bo.it (G.D.); pierandrea.deiaco@unibo.it (P.D.I.); myriam.perrone@unibo.it (A.M.P.); 2Department of Medical and Surgical Sciences, University of Bologna, 40138 Bologna, Italy; camelia.coada@unibo.it (C.A.C.); damiano.paoli4@unibo.it (D.P.); 3Scientific & Medical Department, IGEA S.p.A., 41012 Carpi, Italy; f.deterlizzi@igeamedical.com; 4Department of Biomedical Sciences and Human Oncology, University of Bari, 70121 Bari, Italy; damiani14@alice.it; 5Gynecologic Oncology Unit, Fondazione IRCCS Istituto Nazionale dei Tumori, 20133 Milan, Italy; francesco.raspagliesi@istitutotumori.mi.it (F.R.); bogani.giorgio@gmail.com (G.B.); antonino.ditto@istitutotumori.mi.it (A.D.); 6Radiation Oncology, IRCCS Azienda Ospedaliero-Universitaria di Bologna, 40138 Bologna, Italy; alessio.morganti2@unibo.it

**Keywords:** gynecological cancer, breast cancer, laser therapy, radiotherapy, sexual function, vulvo-vaginal atrophy

## Abstract

**Simple Summary:**

Patients diagnosed with gynecological and breast cancer undergo multimodal treatments leading to estrogen deprivation and vaginal damage in case of radiotherapy, resulting in significant impairments of vulvo-vaginal function. Non-ablative intravaginal CO2 laser is a promising technique for VVA (vulvo-vaginal atrophy) in breast cancer, gynecological and other pelvic cancer survivors. In this study we explored the effectiveness and the long-term effects of repeated cycles of laser therapy. This therapeutic regimen could represent an effective treatment option for patients with limited therapeutic alternatives due to the hormone-sensitivity frequently showed by these cancers.

**Abstract:**

This study aimed to investigate the efficacy and duration of multiple non-ablative intravaginal CO2 laser (V-lase^®^) cycles in breast cancer patients, gynecological and other pelvic cancers previously subjected to multiple oncological treatments. This prospective study enrolled women under the age of 65 years who reported vaginal symptoms. Data on the Vaginal Health Index (VHI), vaginal length (VL), vaginal pain measured using a Visual Analog Scale (VAS), and the Female Sexual Function Index (FSFI) were collected at baseline and before each laser application, and at subsequent follow-up visits. A total of 170 laser applications were performed on 113 women with various types of cancer. Most patients (57.5%) had received radiotherapy-based treatments before receiving laser treatment. Vaginal health parameters and sexual function improved significantly with each laser application. However, a temporary decline in these improvements occurred during the intervals between cycles. Such worsening was reversed with the subsequent cycle in all groups of patients, irrespective of the type of oncological treatments they had undergone. Multiple course vaginal laser therapy showed promising results as a potential treatment for vaginal atrophy in heavily treated gynecological and breast cancer patients, necessitating further research to determine the optimal time interval between cycles to ensure sustained positive effects.

## 1. Introduction

Patients diagnosed with hormone-sensitive cancers, such as breast cancer and gynecological cancers, often undergo multimodal treatment involving surgery, chemotherapy, radiotherapy and antihormonal therapy, either alone or in combination [1,2]. These treatment modalities can lead to estrogen deprivation and vaginal damage in the case of radiotherapy, resulting in various general and local side effects, including vulvo-vaginal atrophy (VVA), which can significantly impair patient’s quality of life [3,4]. Even though treatment options to alleviate these symptoms are available, they are often limited due to contraindications or ineffectiveness of systemic and local hormone replacement therapies, especially after radiotherapy, due to the reduced presence in hormone receptors [5,6,7,8].

In the gynecological field, the use of laser therapy has evolved to encompass both a curative role (for example the treatment of HPV-associated lesions), as well as to improve the signs of vaginal atrophy, emerging as a potential alternative treatment for non-oncological postmenopausal VVA [9,10]. Although the underlying mechanisms of action are not fully understood, laser therapy has shown promise in remodeling the extracellular matrix, increasing collagen production, improving vaginal elasticity, and enhancing hydration of the vaginal mucosa. This process has been reported to alleviate urinary symptoms and enhance sexual intercourse [11,12].

Following the positive results in women with spontaneous menopause [13,14], studies were initiated in people who experienced iatrogenic menopause due to oncological treatments and who were unable to undergo hormonal therapy due to medical contraindications, such as breast [15] and gynecological cancers [16]. Preliminary results have indicated promising results in terms of symptoms relief and overall well-being in these women [17,18,19]. A significant improvement in the Vaginal Health Index (VHI) and a reduction in perceived dyspareunia in 76% of patients after one month of treatment were observed [20] as well as an improvement in urinary symptoms and Female Sexual Function Index (FSFI) scores [21]. However, these effects diminished after more than one year [22].

In addition to hormone deprivation, another important consideration for vaginal health is pelvic radiotherapy, which can cause damage to the vagina and result in local hormonal insensitivity. In a previous study conducted by our team, we explored the use of non-ablative CO_2_ laser (V-Lase^®^ Lasering, Modena, Italy) in women previously treated with radiation therapy for cervical and endometrial cancer. During a six-months follow-up period, these women received three laser treatments, resulting in a gradual improvement in vaginal length (VL) and VHI scores. However, we did not observe a significant improvement in sexual function [23].

Despite significant advances in laser therapy for the treatment of VVA in oncological patients, there are still unanswered questions regarding the long-term effects as well as the sustainability of the results over time. To fill this knowledge gap, our study aimed to evaluate the long-term effects of repeated laser therapy cycles on vaginal physical parameters and sexual function in patients with breast, gynecological and other pelvic cancers. Specifically, we aim to: (i) investigate whether the efficacy of laser therapy increases after multiple treatment cycles in comparison to the initial treatment; (ii) determine the duration of the observed effects on vaginal physical parameters and sexual function; (iii) explore any potential influence of different oncological treatments, such as surgery, chemotherapy or antihormonal therapy, and radiotherapy, on the outcomes of laser therapy.

## 2. Materials and Methods

### 2.1. Study Design

This prospective study was conducted in accordance with the Declaration of Helsinki at the Division of Oncologic Gynecology, Bologna, Italy, between November 2017 and February 2022 and was approved by the local Ethics Committee (registration code 401/2018/OssAOUBo).

Laser therapy cycles were administered to eligible patients as a part of the treatment protocol [23]. A laser cycle consisted of three laser applications, with approximately 30–40 days between each application. The treatment schedule included baseline measurements at T0 (first application), subsequent measurements at T1 and T2 before each laser application, and a follow-up visit at T3 after the final application, which was scheduled with the same time 30–40 days interval. Additional laser cycles were offered to patients who wished for further treatment at six months after completion of the previous cycle (Figure 1).

### 2.2. Laser Procedure

The laser treatment employed in this study utilized a non-ablative intravaginal CO_2_ laser (V-Lase^®^, Modena, Italy) as previously described [23]. Briefly, the procedure consisted of inserting a cylindrical metal probe into the vaginal cavity up to the vaginal dome or the cervix, according to any previous surgery and by placing a graduated metal guide at the level of the vaginal introitus. Laser pulses were generated to treat the entire vaginal wall, with both circumferential and longitudinal movements, including the vaginal introitus and the vestibule. The pulse duration of the laser was 1.5 ms with an interval of 1.5 ms between pulses. The emission time of the laser sequences was 150 ms with an average power of 13 W resulting in an energy density of 1.24 J/cm^2^.

### 2.3. Assessment of Vaginal Physical Parameters and Sexual Function Questionnaire

The evaluation of vaginal physical parameters and sexual function involved multiple assessments conducted at specific time points (labelled as T0, T1, T2, T3 for each time point). Roman numbers were used to indicate the cycle number in which these measurements were taken (I for the first cycle, II for the second cycle and III for the third cycle). Measurements were performed prior to each laser session (T0, T1, T2) and during the follow-up visit (T3). Vaginal Length (VL) [24] was measured to assess any changes during the treatment cycle. The Vaginal Health Index (VHI) [25] was used as a comprehensive measure of vaginal health, evaluating parameters such as elasticity, secretions, pH, epithelial mucous membrane, and hydration. Vaginal pain during laser therapy was assessed using a visual analogue score (VAS) [26] ranging from 0 (no pain) to 10 (maximum pain), allowing patients to indicate their perceived pain intensity (Appendix A). Furthermore, the Female Sexual Function Index (FSFI) [27], a validated questionnaire, was employed to evaluate various domains of sexual function, including desire/libido, arousal, vaginal lubrication, orgasm, global satisfaction or “global quality of life”, and dyspareunia (Appendix A).

### 2.4. Population

The patients were selected from outpatient clinics of the Division of Oncologic Gynecology. Relevant patient data and oncological treatment details were retrieved from available medical records and were reported in an electronic database. The collected information included general data (age, BMI, comorbidity, gynecological and obstetric history) and oncological data (histology, tumor stage, type of oncological treatments and recurrence).

Inclusion criteria were: (i) women under 65 years of age diagnosed with various pelvic or breast cancers; (ii) received uterine surgery or pelvic radiotherapy-based treatments or antihormonal drugs or chemotherapy as their oncological treatment; (iii) experiencing vaginal discomfort (e.g., vaginal dryness or difficulties with sexual intercourse; (iv) written informed consent. Exclusion criteria were: (i) a disease-free follow-up period of less than six months; (ii) diagnosis or suspicion of disease recurrence; (iii) recurrent urinary tract infection or active genital infection; (iv) patients who did not complete the three applications of a laser therapy cycle as well as the questionnaires.

### 2.5. Statistical Analysis

Continuous data were presented as means and standard deviations, while qualitative variables were expressed as absolute numbers and frequencies. Group comparisons were performed using the Student’s t-test, repeated measures ANOVA, and chi-squared tests. All tests were two-sided and a *p*-value < 0.05 was considered for statistical significance. Analyses were performed using GraphPad Prism 8 and NCSS software. Sample size estimation was carried out in R [28] using the *rmcorr* [29] and the *pwrss* [30] packages. Details regarding patients grouping for all the analyses conducted in this study are found in the Appendix A.

## 3. Results

### 3.1. Population Characteristics

A total of 129 women were initially enrolled. However, 16 patients voluntarily withdrew after the first laser application and a further three patients discontinued laser treatment due to disease recurrence. Therefore, a total of 113 patients meeting the selection criteria were included in the final analysis (Figure 1, Table 1). Prior to undergoing laser treatment, more than half of the patients (57.5%) had received radiotherapy-based treatments, while 25.7% of patients had undergone hysterectomy and bilateral adnexectomy and 16.8% of our population had received chemotherapy/anti-hormonal drugs (Appendix A).

### 3.2. Laser Treatment

A total of 170 laser applications were performed (Figure 1). Out of all participants, 69 (61%) underwent a single laser cycle, 31 (27.5%) underwent two laser cycles, and 13 (11.5%) underwent three laser cycles. Notably, patients who received three laser cycles were statistically younger at their first laser treatment (*p* = 0.04) and experienced an earlier onset of menopause due to cancer treatments (*p* = 0.03) (Table 1). The time intervals between laser applications were approximately one month, while the median interval between laser cycles was 12.13 months (IQR = 10.03–17.85) (Appendix A).

### 3.3. Laser Procedure Tolerability

Almost all adverse reactions to laser treatment were predominantly observed during the first cycle, accounting for 89% of all adverse events recorded. A total of 8 patients experiencing adverse reactions, with the majority (63%) occurring after the first laser application. All documented complications were classified as mild, including vulvar and vaginal burning sensation lasting less than 7 days after the procedure (four patients), itching sensation and transient vulvar erythema (three patients), and minor urinary problems such as single of cystitis (two patients). The average VAS score reported during the laser sessions remained consistent and did not vary significantly between treatments. The laser procedures were well-tolerated, with nearly half (48.8%) of the patients reporting a median pain level of 1 (IQR = 0–4) during the applications.

### 3.4. Vaginal Health Parameters and FSFI Score

To evaluate the efficacy of multiple laser applications and their impact on vaginal health, we analyzed the progression of vaginal health parameters across the whole study population. Figure 2A demonstrates a significant improvement in both VHI and VL following each laser application within the same cycle. Specifically, at the initial assessment (I.T0), the average VHI was 12.8 ± 3.83 and VL was 6.36 ± 1.87 cm. These values improved to 16.3 ± 3.9 and 7.38 ± 1.76 cm, respectively, (*p* < 0.001) at the follow-up visit (I.T3). Similar significant improvements were observed during the second and third laser cycles (*p* < 0.01 for both) (Figure 2A; Appendix A). This trend was seen also when examining the three patient groups separately, categorized by the number of treatment cycles they underwent (Appendix A). Furthermore, to assess the improvements achieved with laser therapy, the patients were categorized into two groups based on normal and abnormal values/scores for VHI (>15), VL (≥6), and FSFI (>26.55). It was observed that the proportion of patients who achieved normal gynecological parameters progressively increased when they underwent more laser cycles (Figure 2B; Table 2).

For the FSFI score, although there was evidence of a gradual increase during the three therapy cycles, this trend only reached statistical significance in the first cycle. However, when we categorized the patients into normal and abnormal values based on a score above 26.55 [31], we observed a higher proportion of patients reaching normal levels from baseline (T0) to T3 with each cycle. Specifically, for Cycle I, the proportion increased from 11% to 23%, for Cycle II it increased from 14% to 24%, and for Cycle III it increased from 0% to 33% [23]. Subsequently, we examined the sustainability of these benefits over time. Time intervals between laser cycles were 11.78 months (6–29.6) and 15.4 months (10–20.7) for I and II cycles and for II-III cycles respectively (Appendix A). A decline in both VHI and VL was observed during the approximately one-year intervals between cycles (Figure 2A,B). Specifically, significant decreases in VHI were observed between the last session of the first cycle (I.T3) and the beginning of the second cycle (II.T0) (16.3 ± 3.9 vs. 14.8 ± 3.43; *p* < 0.001), as well as between the last session of the second cycle (II.T3) and the beginning of the third cycle (III.T0) (17.3 ± 4.21 vs. 14.5 ± 3.5; *p* = 0.01). Despite this decrease, VHI remained significantly higher compared to the baseline values (*p* < 0.001 for CII-T0 and *p* = 0.01 for CIII-T0). However, VL did not maintain the improvements and returned towards the initial values (Figure 2A; Table 3; Appendix A), with a significant decline between I.T3 and II.T0 (7.38 ± 1.76 cm vs. 6.47 ± 1.98 cm; *p* < 0.01), as well as between II.T3 and III.T0 (7.15 ± 1.79 cm vs. 6 ± 1.78 cm; *p* < 0.01).

When dividing the study population into two subgroups based on the time interval between the end of the first cycle and the initiation of the second cycle (less than 18 months vs. more than 18 months), a significant worsening of the previously gained VHI values (*p* < 0.01) was observed in the group receiving the second laser cycle later than 18 months after completion of the first cycle (77.9 + 37.3 vs. 41 + 41.9) (Figure 3; Appendix A). A similar trend was observed for VL although not statistically significant (49.2 + 48.4 vs. 34.7 + 47.1; *p* = 0.38). A similar decline in benefits was evident in the FSFI scores. Specifically, the group of patients who initiated the second laser cycle more than 18 months later experienced an average reduction of 66 ± 42.9 points from the initial improvement achieved during the first laser cycle. In contrast, patients who commenced the second laser cycle earlier only lost 33.4 ± 43.5 points (*p* = 0.036).

### 3.5. Evolution of Vaginal and Sexual Parameters Based on the Different Oncological Treatments

Next, we sought to assess whether the effectiveness of the laser cycles could be influenced by the specific oncological treatments the women had undergone previously. To achieve this, we categorized the patients into three groups based on their previous oncological treatments: those who had received radiotherapy-based treatments (N = 65), those who had undergone hysterectomy and bilateral adnexectomy (N = 29), and those who had received chemotherapy and/or antihormone therapy (N = 19) (Table 3; Appendix A).

Considering only the patients who underwent radiotherapy (N = 65), we consistently observed lower values of VHI and VL throughout the entire duration of laser therapy compared to the other subgroups (Figure 4; Table 3). However, significant improvements were observed in these patients as well, for both VHI (*p* < 0.001, *p* = 0.02, *p* < 0.001 for cycles I, II, and III, respectively) and VL (*p* < 0.001, *p* < 0.001, and *p* = 0.01 for cycles I, II, and III, respectively) (Figure 4; Table 3). Regarding FSFI, a significant increase was observed during the first laser cycle (*p* < 0.001), but this trend was not maintained in subsequent cycles (Figure 4; Table 3). Importantly, there was no significant correlation between the length of time after completion of radiotherapy and the effectiveness of laser treatment (Appendix A).

In the subgroup of patients who underwent hysterectomy and bilateral adnexectomy (N = 29), there was a significant increasing trend observed in both VHI (*p* < 0.001) and VL (*p* < 0.001) during the first laser cycle. Although not statistically significant, a similar increasing trend was observed in both physical parameters during the second and third cycles (Figure 3; Table 3). In the subgroup of patients receiving chemotherapy and/or antihormone therapy (N = 19), a significant increasing trend was observed in VHI values during the second laser cycle (*p* < 0.05). However, for the remaining cycles, although there was an increasing trend in both VHI and VL, these changes did not reach statistical significance. When considering the FSFI score, the results did not consistently align with the changes observed in the physical parameters (Figure 3; Table 3).

## 4. Discussion

In this study, we investigated the effectiveness of laser therapy in improving vaginal health parameters in a cohort of 113 women who received various cancer treatments, including radiotherapy-based treatments. Despite the challenges posed by previous cancer cures and their negative impact on vaginal health [32,33], laser therapy demonstrated promising results in improving physical parameters, such as VHI and VL. Although the results were not fully sustained over the course of a year, the possibility of obtaining the same improvements with the addition of another laser cycle was seen (Figure 3).

In summary, we observed significant improvements in both VHI and VL following each laser application and across all three laser cycles (Figure 2A). However, it is important to note that there was a decline in VHI and VL values during the intervals between the cycles. This decline was significant in the case of patients who began the second laser cycle more than 18 months later, suggesting the need for regular laser applications to maintain the benefits achieved. This finding emphasizes the importance of adherence to the recommended treatment schedule to sustain positive outcomes. By categorizing patients into normal and abnormal groups based on predefined normal values [25,31,34], we observed a progressive increase in the proportion of patients achieving normal vaginal parameters with each cycle of laser therapy. The safety profile of the procedure was highly favorable, with almost all adverse reactions observed in the first cycle and classified as mild. This underscores the excellent tolerability of laser therapy.

When analyzing the patient subgroups according to the type of cancer treatment received, we observed that patients who underwent radiotherapy had lower baseline VHI and VL scores (Figure 3; Table 3). However, significant improvements in vaginal health were also observed in these patients, indicating that laser therapy can effectively improve vaginal health even in this context. These positive results are encouraging as these patients are particularly at risk of developing further stenosis and worsening of their condition due to the pathological collagen production and scarring of the vaginal tissue. Albeit not statistically significant, a slightly higher proportion of patients in this subgroup underwent a third laser cycle (Figure 3; Table 1). This may suggest that the perceived improvement after laser treatment was higher in these patients than in the two other subgroups, motivating them to continue the treatment. It is well known that patients previously treated with pelvic radiotherapy-based treatments often experience vaginal stenosis due to fibrosis that affects the mucosal and the submucosal layer of the vaginal walls [23,35]. It is also well known that vaginal laser therapy leads to tissue repair and remodeling [10] in atrophic and fibrotic tissues. Further studies focusing on vaginal biopsies are required to explore the relationship between vaginal fibrosis and the response to laser therapy in radio-treated patients.

Despite the significant improvements in physical parameters, we did not observe consistent changes in FSFI, further confirming our previous results [23]. This discrepancy suggests that factors beyond the physical aspects, such as psychological factors or the presence of a stable partner, may play a significant role in normal sexual function [36,37,38].

Our study reinforces the idea that treatments for gynecological tumors affecting sexual hormones have a marked impact on vaginal health, as revealed by the poor baseline parameters of our patients (Appendix A). Despite the positive results, it is worth noting that a significant proportion of women did not reach normal values at the end of all cycles, namely 25% for VHI and 33% for VL. The possibility of improving outcomes in these patients through multiple subsequent laser applications with a shorter interval between cycles should be further explored. Additionally, our study revealed that patients who underwent radiotherapy experienced poorer outcomes compared to those receiving other treatment modalities. This disparity can likely be attributed to the lower baseline values of the parameters in this patient subgroup (Figure 3; Table 3). However, within this subgroup, there was a more pronounced improvement in vaginal health parameters (Figure 3; Table 3). This is evident from the observed average increase in VHI of 43.8% and VL of 26.1% from I.T0 to I.T3. These findings are particularly noteworthy because they indicate positive outcomes in a subgroup of patients who have limited chances of improvement with hormonal therapies [5], further highlighting the potential efficacy of laser treatment. These results further support our previous findings, showing an average improvement in VHI of 57% and VL of 28% after a single cycle of laser therapy [23]. Just over 11% of the patients chose to complete a third laser cycle. While information regarding the specific reasons behind this choice was not systematically documented, it is reasonable to assume that the discontinuation was not due to the lack of beneficial effects. This is supported by the fact that all the patient groups had similar results in the cycles they received.

Vaginal dilators are commonly recommended as an alternative method to improve vaginal compliance, particularly for oncological patients in the recovery phase [39]. However, it is worth mentioning that laser treatment offers several advantages over the use of vaginal dilators, such as significantly shorter application times and reduced reliance on patients’ willingness [40]. Moreover, there is a lack of high-quality studies providing a direct comparison between laser treatments and other techniques, such as vaginal dilators or hormonal treatments, as recently highlighted by a Cochrane review [40]. Furthermore, our findings suggest that the efficacy of laser therapy is not negatively impacted by the time elapsed since the completion of radiotherapy. This implies that laser therapy can be initiated even later, without compromising its effectiveness. These promising results challenge previous assumptions [41] that delaying the initiation of vaginal rehabilitation may hinder potential improvements. To the best of our knowledge, this is the first study to comprehensively evaluate the effects of vaginal rehabilitation not only immediately after the completion of radiotherapy-based treatments but also at later time points.

The present study has several strengths such as including a relatively large cohort of over 100 patients as well as being the first study to investigate the feasibility and efficacy of multiple cycles of vaginal laser treatments in gynecological and breast cancer patients. This provides valuable insights into the potential benefits of this treatment for improving vaginal health in this specific population. Nonetheless, a notable limitation arises from the relatively small number of patients who successfully completed all three laser cycles. This limitation posed challenges for conducting subgroup analyses and fully comprehending the durability of the laser’s effects over time. Thus, future studies with larger sample sizes are needed to validate and confirm our results and to optimize the use of laser treatment in clinical practice. Furthermore, this study did not focus on the mechanisms through which laser therapy acts on estrogen-deprived vaginal tissue, as no vaginal biopsies were performed before and after the laser treatment. Further studies are needed to explore this important aspect. Moreover, it would be worth investigating the effectiveness of laser therapy compared to other rehabilitation techniques such as the use of vaginal dilators.

## 5. Conclusions

Our study provides valuable insights into the efficacy and safety of laser therapy for improving vaginal health in oncological women. Laser therapy can lead to significant improvements in vaginal health parameters, including patients who received radiotherapy. However, regular laser therapy sessions may be necessary to sustain and enhance these improvements.

## Figures and Tables

**Figure 1 cancers-16-02722-f001:**
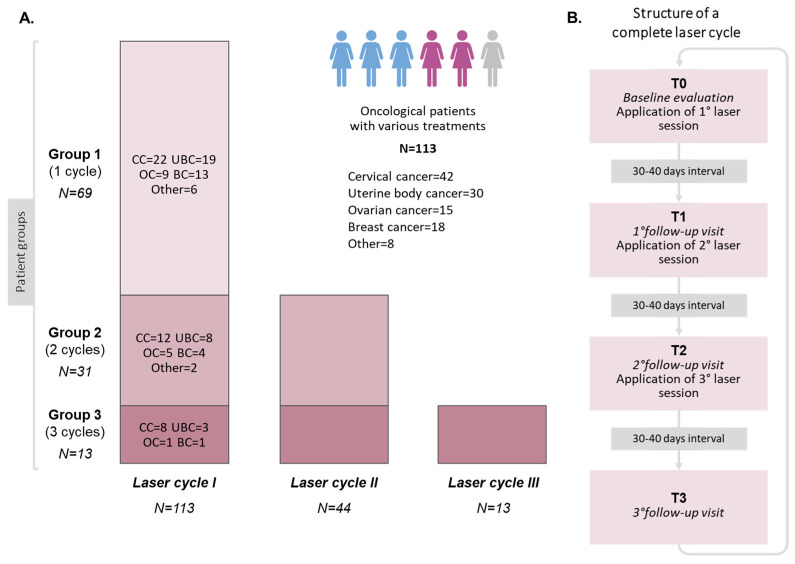
(**A**). Flow chart presenting the three patient groups based on the number of laser cycles they received. (**B**). Structure of a complete laser cycle comprising three laser sessions at a 30–40-days interval. N: number of patients; CC: Cervical cancer; UBC: Uterine body cancer; OC: Ovarian cancer; BC: Breast cancer; other: other cancers i.e., rectal cancer (N = 5), Bartholin glands tumors (N = 3).

**Figure 2 cancers-16-02722-f002:**
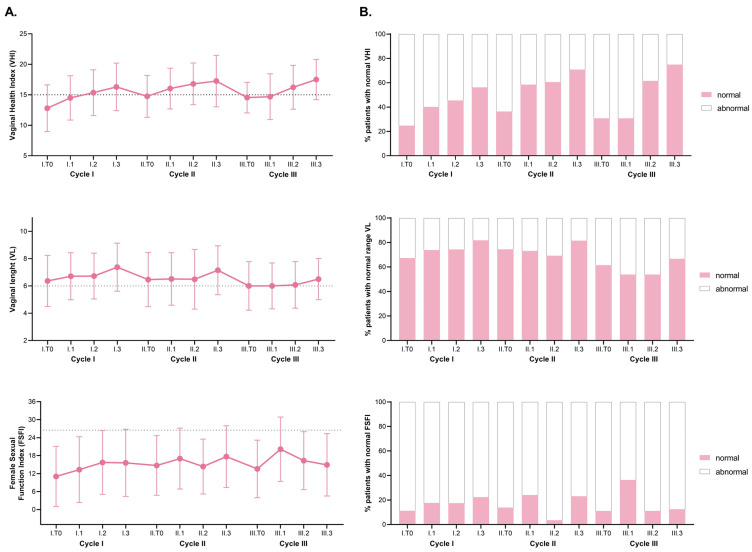
(**A**). Evolution of gynecological parameters during vaginal laser treatment in oncological patients. Dotted lines represent the normal thresholds for VHI (>15), VL (≥6 cm) and FSFI (>26.55), respectively. (**B**). Proportion of patients reaching normal values for VHI, VL and FSFI after the laser sessions. VHI: Vaginal Health Index, VL: vaginal length, FSFI: Female Sexual Function Index.

**Figure 3 cancers-16-02722-f003:**
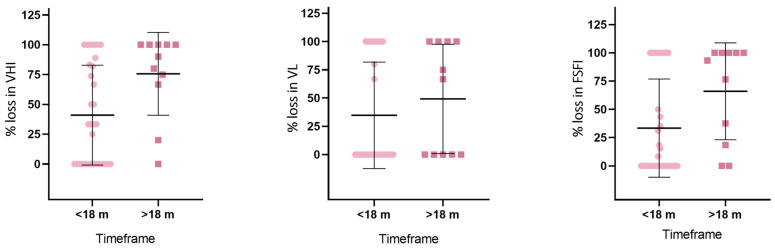
Loss of improvements in the vaginal parameters and function based on the gap length between cycles. VHI: Vaginal Health Index, VL: vaginal length, FSFI: Female Sexual Function Index.

**Figure 4 cancers-16-02722-f004:**
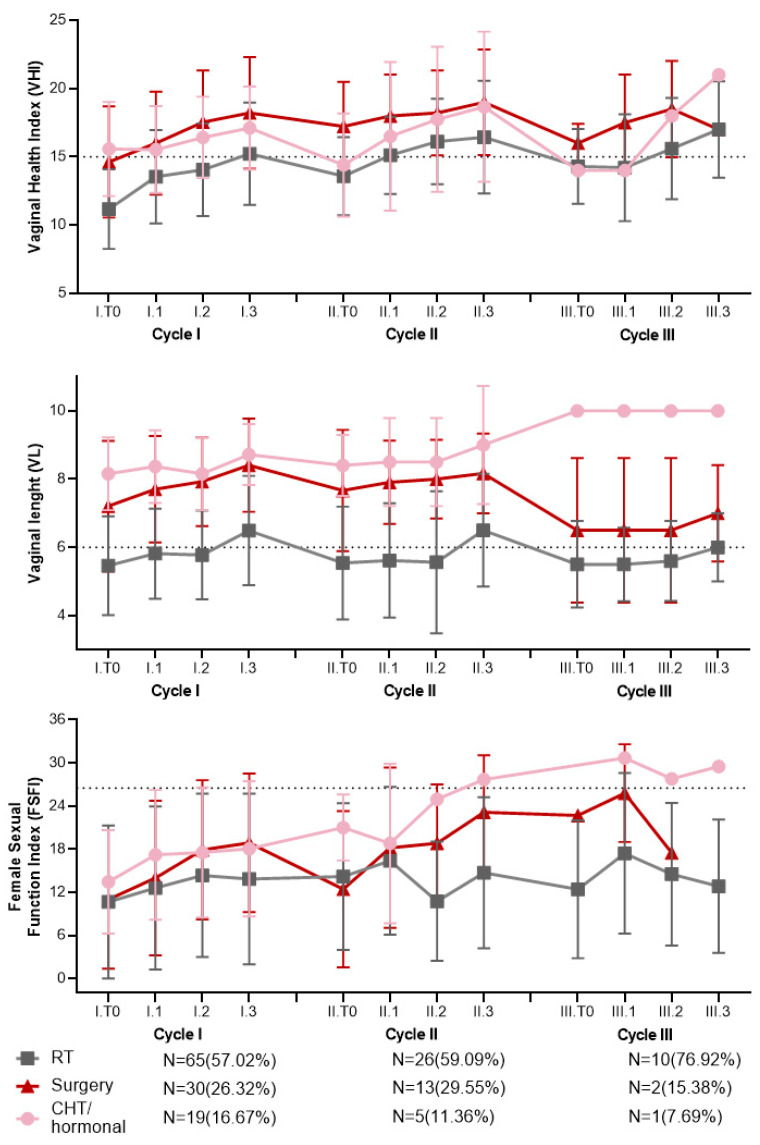
Evolution of VHI and VL values during 1st and 2nd cycle of laser treatment in subgroups of cancer treatment. RT: radiotherapy-based treatment; CHT: chemotherapy; N: number of patients.

**Table 1 cancers-16-02722-t001:** General characteristics of patients receiving 1, 2 or 3 laser cycles. N: number of patients; SD: standard deviation; BMI: body mass index; N/A: not available.

			Patients Group. N(%)/Mean ± SD			
**Variable**		**Total** N = 113	**1 Cycle** N = 69(61.1%)	**2 Cycles** N = 31(27.4%)	**3 Cycles** N = 13(11.5%)	***p*-Value**
**Age at oncological diagnosis**		46.6 ± 8.5	48 ± 9	45.47 ± 6.85	44.57 ± 8.11	0.5
**Age at first laser**		50.5 ± 8.3	51.54 ± 8.97	50.06 ± 6.67	45.69 ± 6.99	0.06
**Age group**	**≤45**	35(31)	18(26.1)	9(29)	8(61.5)	**0.04**
	**>45**	78(69)	51(73.9)	22(71)	5(38.5)	
**Age at menopause**		45.2 ± 6.3	45.48 ± 6.62	45.62 ± 5.39	43.43 ± 5.79	0.38
**Early menopause < 45 years**	**No**	58(52.3)	40(59)	15(54.8)	3(23.1)	0.07
	**Yes**	53(47.7)	29(41)	14(45.2)	10(76.9)	
	*N/A*	*2*				
**Premature menopause < 40 years**	**No**	94(84.7)	59(85.5)	27(93.5)	8(61.5)	**0.03**
	**Yes**	17(15.3)	10(14.5)	2(6.5)	5(38.5)	
	*N/A*	*2*				
**BMI (kg/m^2^)**		23.49 ± 4.4	23.71 ± 4.9	23.63 ± 3.42	21.97 ± 4.43	0.42
**BMI category**	**Normal**	100(89.3)	58(84.1)	30(96.7)	12(92.3)	0.22
	**Overweight**	12(10.7)	10(15.9)	1(3.3)	1(7.7)	
**Parity**	**Nulliparous**	43(39.1)	24(36.4)	14(45.2)	5(38.5)	0.42
	**Parous**	55(50)	32(48.5)	15(48.4)	8(61.5)	
	**Cesarean section**	12(10.9)	10(15.1)	2(6.5)	-	
	*N/A*	*3*				
**Tumor type at diagnosis**	**Cervix**	42(37.2)	22(31.9)	12(38.7)	8(61.5)	0.84
	**Uterus**	30(26.5)	19(27.6)	8(25.8)	3(23.1)	
	**Rectal**	5(4.4)	4(5.8)	1(3.2)	-	
	**Ovarian**	15(13.3)	9(13)	5(16.1)	1(7.7)	
	**Breast**	18(15.9)	13(18.8)	4(13)	1(7.7)	
	**Other**	3(2.7)	2(2.9)	1(3.2)	-	
**Stage**	**I + II**	86(76.1)	52(75.4)	27(87.1)	12(92.3)	0.11
	**III + IV**	25(23.9)	20(24.6)	3(12.9)	2(7.7)	
	*N/A*	*2*				
**Radiotherapy and/or Brachytherapy**		65(57.6)	39(56.5)	16(51.6)	10(76.9)	0.36
**Hysterectomy and Bilateral adnexectomy**		30(26.6)	16(23.2)	11(35.5)	2(15.4)	
**Chemotherapy and anti-hormonal therapy**		19(16.8)	14(20.3)	4(12.9)	1(7.7)	

**Table 2 cancers-16-02722-t002:** Proportion of women reaching normal VHI, VL and FSFI values after laser cycle sessions (related to Figure 2B).

Cycle Number	Laser Application	Number of Patients with VHI > 15 N/Total(%)	Number of Patients with VL ≥ 6 cm N/Total(%)	Number of Patients with FSFI > 26.55 N/Total(%)
**I**	**T0 Before Treatment Initiation**	28/113(24.78)	76/113(67.26)	10/89(11.24)
	**T1**	45/112(40.18)	82/111(73.87)	15/85(17.65)
	**T2**	50/110(45.46)	81/109(74.31)	15/86(17.44)
	**T3**	54/96(56.25)	85/104(81.73)	17/76(22.37)
**II**	**T0 Before Cycle II**	16/44(36.37)	32/43(74.42)	5/36(13.89)
	**T1**	24/41(58.54)	30/41(73.17)	7/29(24.14)
	**T2**	23/38(60.53)	27/39(69.23)	1/28(3.57)
	**T3**	17/24(70.83)	22/27(81.48)	3/13(23.08)
**III**	**T0 Before Cycle III**	4/13(30.77)	8/13(61.54)	1/9(11.11)
	**T1**	4/13(30.77)	7/13(53.85)	4/11(36.36)
	**T2**	8/13(61.54)	7/13(53.85)	1/9(11.11)
	**T3**	6/8(75)	8/12(66.67)	1/8(12.5)

VHI: Vaginal Health Index, VL: vaginal length, FSFI: Female Sexual Function Index.

**Table 3 cancers-16-02722-t003:** Descriptive statistics showing the evolution of the VHI, LV and FSFI parameters during the laser therapy sessions in patients who received radiotherapy-based treatments (radiotherapy, brachytherapy), surgery and received chemotherapy and/or hormonal therapy (related to Figure 3).

Radiotherapy-Based Treatments (Radiotherapy, Brachytherapy)							
		VHI (Points)		LV (cm)		FSFI (Points)	
Cycle Number	Laser Session	Mean ± SD	*p*-Value	Mean ± SD	*p*-Value	Mean ± SD	*p*-Value
**I** N = 65	**T0**	11.2 ± 2.92	**<0.001**	5.46 ± 1.45	**<0.001**	10.7 ± 10.6	**0.05**
	**T1**	13.5 ± 3.43		5.82 ± 1.32		12.6 ± 11.3	
	**T2**	14 ± 3.4		5.78 ± 1.3		14.4 ± 11.4	
	**T3**	15.2 ± 3.75		6.49 ± 1.6		13.9 ± 11.9	
**II** N = 26	**T0**	13.6 ± 2.86	**0.02**	5.54 ± 1.66	**<0.001**	14.2 ± 10.2	0.62
	**T1**	15.1 ± 2.83		5.62 ± 1.68		16.8 ± 10.6	
	**T2**	16.1 ± 3.14		5.56 ± 2.08		10.8 ± 8.3	
	**T3**	16.4 ± 4.13		6.5 ± 1.65		14.7 ± 10.5	
**III** N = 10	**T0**	14.3 ± 2.75	**<0.001**	5.5 ± 1.27	**0.01**	12.4 ± 9.59	0.88
	**T1**	14.2 ± 3.91		5.5 ± 1.08		17.4 ± 11.2	
	**T2**	15.6 ± 3.72		5.6 ± 1.17		14 ± 10.8	
	**T3**	17 ± 3.52		6 ± 1		12.8 ± 9.27	
**Surgery**							
		**VHI** (Points)		**LV** (cm)		**FSFI** (Points)	
**Cycle Number**	**Laser Session**	**Mean ± SD**	***p*-Value**	**Mean ± SD**	***p*-Value**	**Mean ± SD**	***p*-Value**
**I** N = 29	**T0**	16 ± 3.78	**<0.001**	7.21 ± 1.92	**<0.001**	11 ± 9.63	**0.01**
	**T1**	14.6 ± 4.07		7.7 ± 1.56		14 ± 10.8	
	**T2**	17.5 ± 3.8		7.93 ± 1.3		17.9 ± 9.67	
	**T3**	18.2 ± 4.09		8.41 ± 1.37		18.9 ± 9.64	
**II** N = 13	**T0**	17.2 ± 3.27	0.84	7.67 ± 1.78	0.89	12.4 ± 10.9	0.12
	**T1**	18 ± 3.03		7.91 ± 1.22		18.2 ± 11.1	
	**T2**	18.2 ± 3.11		8 ± 1.16		18.8 ± 8.21	
	**T3**	19 ± 3.87		8.17 ± 1.17		23.1 ± 7.93	
**III** N = 2	**T0**	16 ± 1.41	n/a	6.5 ± 2.12	n/a	19.8 ± 4.03	n/a
	**T1**	17.5 ± 3.54		6.5 ± 2.12		25.8 ± 6.79	
	**T2**	18.5 ± 3.54		6.5 ± 2.12		22.8 ± 7.42	
	**T3**	20 ± 4.24		7 ± 1.41		24.5 ± 10.6	
**Chemotherapy and/or Hormonal Therapy**							
		**VHI (Points)**		**LV (cm)**		**FSFI (Points)**	
**Cycle Number**	**Laser Session**	**Mean ± SD**	***p*-Value**	**Mean ± SD**	***p*-Value**	**Mean ± SD**	***p*-Value**
**I** N = 19	**T0**	15.6 ± 3.45	0.08	8.16 ± 1.07	0.16	13.5 ± 7.17	**0.47**
	**T1**	15.5 ± 3.19		8.37 ± 1.06		17.2 ± 9.01	
	**T2**	16.4 ± 2.99		8.16 ± 1.07		17.6 ± 9.05	
	**T3**	17.1 ± 3.05		8.72 ± 0.89		18.1 ± 9.41	
**II** N = 5	**T0**	14.4 ± 3.78	**0.05**	8.4 ± 0.894	0.28	21 ± 4.59	0.67
	**T1**	16.5 ± 5.45		8.5 ± 1.29		18.4 ± 9.09	
	**T2**	17.8 ± 5.32		8.5 ± 1.29		19.6 ± 11.3	
	**T3**	18.7 ± 5.51		9 ± 1.73		24.1 ± 6.21	
**III** N = 1	**T0**	14	n/a	10	n/a	27	n/a
	**T1**	14		10		30.7	
	**T2**	18		10		27.8	
	**T3**	21		10		29.5	

## Data Availability

Data available on reasonable request from the authors.

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
