# Peer review of "Laser Therapy in Heavily Treated Oncological Patients Improves Vaginal Health Parameters"

_cancers, 2024, doi:10.3390/cancers16152722_

Round 1

Reviewer 1 Report

Comments and Suggestions for Authors

Dear authors, thank for this interesting paper. However it has many flaws to be discussed before considering it for publication. 

Line 17: not damage, it is only atrophy from estrogen deprivation. Please amend, same inline 46.  Damage should’ve specified only for patients city vaginal irradiation

Line 56: many studies have shown that laser vaginal therapy ( please avoid using the term vaginal rejuvation since it is mesleading) is ineffective in improving post menopausal symptoms (refer to 10.1001/jamanetworkopen.2022.55706)

Please always use high evidence when talking about patients, especially in a paper for oncological patients.

Line 78: I wonder the long follow up if the follow up visit is scheduled after 30-40 days after last application

In methods please specify technical setting of laser machine

I would specify in results two important aspects: median time from menopause to laser treatment and how many patients went in menopause due to oncological treatment.

I wonder also the physiological base of application of laser: especially in women who underwent brachitherpah or vaginal irradiation, they had vaginal fibrosis and stenosis. A further collagen production and remodelling could possibly make the vagine even more fibrotic. This should be discussed in details.

Main limitation of this study is the absence of a sham or control group. 

Comments on the Quality of English Language

Minor to moderate

Author Response

Dear authors, thank for this interesting paper. However it has many flaws to be discussed before considering it for publication. 

R: Thank you for the constructive comments. We went through each suggestion and made changes accordingly.

Line 17: not damage, it is only atrophy from estrogen deprivation. Please amend, same inline 46.  Damage should’ve specified only for patients city vaginal irradiation

R: Thank you for pointing this out. We added “in case of radiotherapy treatment” in the sentence to clarify this aspect. (page 1 line 17 and page 2 line 47)

Line 56: many studies have shown that laser vaginal therapy ( please avoid using the term vaginal rejuvation since it is mesleading) is ineffective in improving post menopausal symptoms (refer to 10.1001/jamanetworkopen.2022.55706)

Please always use high evidence when talking about patients, especially in a paper for oncological patients.

R: Thank you for this suggestion and the useful reference. We cited it in the manuscript. We also removed the term “vaginal rejuvenation” as requested. (page 2 line 57)

Line 78: I wonder the long follow up if the follow up visit is scheduled after 30-40 days after last application

R: Thank you for highlighting this issue. We confirm that the time between sessions was 30-40 days. The time between cycles was longer as the patients returned as they felt necessary without any suggested timeframe. They were only requested to come 30-40 days after the last laser session of the cycle for evaluation and measurements. However, for the initiation of a whole new cycle, some patients returned and requested one even after more than 18 months (please see Figure 3). (page 2 line 97)

In methods please specify technical setting of laser machine

R: We added details regarding the technical settings: “The pulse duration of the laser was 1.5 ms with an interval of 1.5 ms between pulses. The emission time of the laser sequences was 150 ms with an average power of 13 W resulting in an energy density of 1.24 J/cm2.” (page 3 lines 107-109)

I would specify in results two important aspects: median time from menopause to laser treatment and how many patients went in menopause due to oncological treatment.

R: Thank you for the comment. Unfortunately, we have not collected this data. It is difficult to obtain a precise interval between menopause and the start of laser therapy in terms of months, especially considering the difficulty of determining the exact month in which the woman went into menopause. Often patients would recall the year but not the month so conducting such an analysis would be severely biased and unprecise. Instead, what we were able to do was divide our patients based on the onset of menopause, whether it was early or premature menopause. (please see supplementary table 1).

I wonder also the physiological base of application of laser: especially in women who underwent brachitherpah or vaginal irradiation, they had vaginal fibrosis and stenosis. A further collagen production and remodelling could possibly make the vagine even more fibrotic. This should be discussed in details.

R: Thank you for this question. Indeed, this is an interesting aspect especially since the mechanisms of atrophy/damage are different based on the specific oncological treatment they received. The aim of our study was to see if laser applications would improve the vaginal wellbeing of oncological patients, so we did not evaluate through biopsies the level of collagen production, the baseline state of the tissue and its evolution throughout the treatment. We conducted this study based on functional evaluation and repeated measurement to document the evolution of the parameters. We do not know the mechanism of action of the laser and further studies are needed to explore this aspect. We added this aspect to the limitations of this study: “Furthermore, this study did not focus on the mechanisms through which laser therapy acts on estrogen-deprived vaginal tissue, as no vaginal biopsies were performed before and after the laser treatment” (page 12 line 360-362).

Nevertheless, our results showed a benefit of the laser applications even in patients receiving radiotherapy-based treatments, so we believe that these encouraging results point towards a remodelling that does not lead to more fibrosis and further stenosis. However, we agree that these findings need further validation on larger cohorts. We commented this issue in the discussion section: “When analyzing the patient subgroups categorized by the type of cancer treatment received, we observed that patients who underwent radiotherapy had lower baseline VHI and VL values (Figure 3; Table 3). However, significant improvements in vaginal health were observed in these patients as well, indicating that laser therapy can effectively improve vaginal health even in this context. These positive results are encouraging as these patients are particularly at risk of developing further stenosis and aggravation of their condition due to the pathological collagen production and scarring of the vaginal tissue. “ (page 11 lines 298-304)

Main limitation of this study is the absence of a sham or control group. 

R: We discussed the lack of high-quality studies comparing the efficacy of various techniques aiming to improve vaginal parameters in oncological women in the limitations section of the discussion. We believe that multicentric studies are required to fill in this knowledge gap. “Vaginal dilators are commonly recommended as an alternative method to improve vaginal compliance, particularly for oncological patients in the recovery phase [38]. However, it is worth mentioning that laser treatment offers several advantages over the use of vaginal dilators such as significantly shorter application times and reduced reliance on patients’ willingness [39]. Moreover, there is a lack of high-quality studies providing a direct comparison between laser treatments and other techniques, such as vaginal dilators or hormonal treatments, as recently highlighted by a Cochrane review [39].” (page 12 lines 337-343)

Reviewer 2 Report

Comments and Suggestions for Authors

Thank you for giving me an apportunitiy to review this comprehensive article.

The major context is well-understood. However, there are a few minor points to enhance the quality of the manuscript.

  • Sample Size and Duration: Although the study includes a relatively large cohort, the sample size for some subgroups, particularly those who completed multiple laser cycles, is small. This small sample size limits the statistical power and the ability to draw definitive conclusions about the long-term effects and efficacy of the treatments​(cancers-3095168-peer-re…)​.

  • Lack of Control Group: The absence of a control group or comparison with other treatment modalities such as vaginal dilators or hormonal treatments limits the ability to definitively attribute observed improvements to the laser therapy alone. Comparative studies would provide more robust evidence of the treatment's efficacy​(cancers-3095168-peer-re…)​.

  • Potential Selection Bias: There might be selection bias in the patient population, as those who opted for and completed the laser treatments could be different from those who did not in terms of severity of symptoms, health status, or other factors. This could influence the study's outcomes and limit the applicability of the results to a broader patient population​

Author Response

Thank you for giving me an apportunitiy to review this comprehensive article.

The major context is well-understood. However, there are a few minor points to enhance the quality of the manuscript.

R: Thank you for the useful comments that gave us the opportunity to further explain and clarify important points of our work. Please find our point-by-point responses to your comments.

Sample Size and Duration: Although the study includes a relatively large cohort, the sample size for some subgroups, particularly those who completed multiple laser cycles, is small. This small sample size limits the statistical power and the ability to draw definitive conclusions about the long-term effects and efficacy of the treatments​(cancers-3095168-peer-re…)​.

R: We fully agree with the reviewer that this is a major limitation of the study and the generalize the conclusions is limited especially in the case of the patients treated with chemotherapy/hormonal therapy and surgery. In fact, we addressed it in the limitations of the study. We discussed this issue multiple times and even considered presenting the subgroup analysis as supplementary data, leaving only the pooled analysis in the main manuscript. However, we decided that a full and thorough presentation of the results with accent on the number of cases in each group (we paid attention to add the number of cases in all the tables and figures to draw attention to this limit) would allow a transparent presentation of the overall encouraging results of this therapy. We sincerely hope that other research groups will consider conducting similar studies and validate our findings, especially because our sample size analysis revealed a number of 8 patients to offer a sufficient power for the data analysis, which was achieved by the group who received radiotherapy-based treatments (supplementary methods). We discussed these issues in the limitations of the study: “Nonetheless, a notable limitation arises from the relatively small number of patients who successfully completed all three laser cycles. This limitation posed challenges for conducting subgroup analyses and fully comprehending the durability of the laser's effects over time. Thus, future studies with larger sample sizes are needed to validate and confirm our results and to optimize the use of laser treatment in clinical practice.“ (lines 358-363)

Lack of Control Group: The absence of a control group or comparison with other treatment modalities such as vaginal dilators or hormonal treatments limits the ability to definitively attribute observed improvements to the laser therapy alone. Comparative studies would provide more robust evidence of the treatment's efficacy​(cancers-3095168-peer-re…)​.

R: Indeed, the next step would be to compare the efficacy of the laser applications with other treatments. However, the design of the current study was that of a phase II trial in which the aim was to establish the treatment scheme, the long-term effects and the influence of various oncological treatments. As far as the ability to definitively attribute observed improvements to the laser therapy alone, this aspect can be mainly controlled by the repeated measurements done in each patient. Since the improvements were constantly seen at each laser session and with each cycle, we can safely affirm that they are due to the laser treatment. Especially because in the time gap between the laser cycles (which was longer; median time around a year) the patients presented a loss of the beneficial effects obtained during the laser cycle and returned to values closed to those at the starting point. Of course, ideally one would like to have also a control group in which patients with the same characteristics as the study group are measured constantly to prove the link with the laser, or even better, a crossover trial in which the same patients would receive the treatment for some time and then would not. Since these studies are harder to perform, they need to be built on encouraging preliminary data which we hope that our study offers. The need for further studies is addressed at the end of the discussion section: “Moreover, it would be worth investigating the effectiveness of laser therapy compared to other rehabilitation techniques such as the use of vaginal dilators.” (lines 371-372)

Potential Selection Bias: There might be selection bias in the patient population, as those who opted for and completed the laser treatments could be different from those who did not in terms of severity of symptoms, health status, or other factors. This could influence the study's outcomes and limit the applicability of the results to a broader patient population​

R: We agree that this might be an issue since the patients were able to choose freely whether to continue with the cycle or not, we could not control the dropout rates. While some patients did not continue the cycles due to the recurrence of the primary disease, others were not documented. Sinse this data was non systematically gathered; we could not report it in the paper. However, what we can say is that the patients that discontinued the laser cycles did not do so because they did not obtain good results in the previous cycle. We show this by analyzing the 3 groups of patients (G1: received 1 cycle vs G2: received 2 cycles vs G3: received 3 cycles). As can be seen from Supplementary Figure1, all 3 groups had similar improvements during the cycles, suggesting that the reason for discontinuation was not related to the treatment itself. We discussed this in lines 335-339: “A little over 11% of the patients chose to complete a third lase cycle. While information regarding the specific reasons behind this choice was not systematically documented, it is reasonable to assume that the discontinuation was not due to the lack of beneficial effects. This is supported by the fact that all the patient groups had similar results in the cycles the received.”

Reviewer 3 Report

Comments and Suggestions for Authors

Thank you for the opportunity to review this article. Vaginal health is an important aspect following oncology treatment and can have a negative impact on quality of life.

As hormonal treatment is often not an option for these patients, laser seems to be a good alternative.

Line 50: Description of VVA and treatment options, article by N Kovacevic et al: Modern approach to the management of genitourinary syndrome in women with gynecological malignancies

doi: 10.2478/raon-2023-0038 , published in 2023 could be mentioned in the introduction

Line 117: VAS scale...an explanation should be added that 0 stands for minimal pain and 10 for maximum pain

Line 129: Inclusion criteria: It is written that women with gynecologic cancer and breast cancer were included in the study? Why does it say different types of pelvic cancer? Were women with colorectal cancer or a retroperitoneal pelvic tumor of non-gynecological origin also included? A paraphrase or detailed explanation is required

Line 130: pelvic radiotherapy...this means EBRT or intravaginal brachyradiotgherapy or both

Line 156: Figure 1 A is not self-explanatory. Perhaps the groups should be placed on the x-axis. 8 patients had 'other' cancers... it is never explained what other cancers are? Gynecologic or non-gynecologic origin? Should these 8 cases be excluded from the statistics?

Figure 1B: The intervals between aplications were 30-40 days... the figure only mentions 30 days. The data do not match

Line 162: A total of 171 laser aplications were carried out. If I look at Figure 1A and count the laser cycle I + II + III, I get 170 applications... this data does not match the figure.

Line 270:Chemio/hormonal...mis-typed. 

Author Response

Thank you for the opportunity to review this article. Vaginal health is an important aspect following oncology treatment and can have a negative impact on quality of life. As hormonal treatment is often not an option for these patients, laser seems to be a good alternative.

R: Thank you for the constructive feedback. We went through all the comments and made adjustments to the manuscript accordingly.

Line 50: Description of VVA and treatment options, article by N Kovacevic et al: Modern approach to the management of genitourinary syndrome in women with gynecological malignancies

doi: 10.2478/raon-2023-0038 , published in 2023 could be mentioned in the introduction

R: Thank you for this suggestion. We reviewed the article and added it in the references.

Line 117: VAS scale...an explanation should be added that 0 stands for minimal pain and 10 for maximum pain

R: Thank you for this suggestion. Details regarding the methods are in the supplementary files. This information was also added in the main text: “Vaginal pain during laser therapy was assessed using a visual analogue score (VAS) [25] ranging from 0 (no pain) to 10 (maximum pain), allowing patients to indicate their perceived pain intensity (Supplementary methods). “ (page 3 line 120)

Line 129: Inclusion criteria: It is written that women with gynecologic cancer and breast cancer were included in the study? Why does it say different types of pelvic cancer? Were women with colorectal cancer or a retroperitoneal pelvic tumor of non-gynecological origin also included? A paraphrase or detailed explanation is required

R: We added this info in the caption of Figure 1.

Line 130: pelvic radiotherapy...this means EBRT or intravaginal brachyradiotgherapy or both

R: Thank you for the question. Patients received Radiotherapy and/or Brachytherapy. To avoid confusion, we always referred to these treatments and Radiotherapy-based. Adjustments have been made throughout the manuscript.

Line 156: Figure 1 A is not self-explanatory. Perhaps the groups should be placed on the x-axis. 8 patients had 'other' cancers... it is never explained what other cancers are? Gynecologic or non-gynecologic origin? Should these 8 cases be excluded from the statistics?

R: Thank you for this comment. We adjusted Figure 1 and hope that it is clear now.

Figure 1B: The intervals between aplications were 30-40 days... the figure only mentions 30 days. The data do not match

R: Thank you for pointing this out. Indeed, the target was to have the visits after 30 days and the figure was made with this in mind, but this exact precision is not always possible in clinical practice and patients were sometimes scheduled with a delay of a few days. We corrected the figure to be more precise and added 30-40 days.

Line 162: A total of 171 laser aplications were carried out. If I look at Figure 1A and count the laser cycle I + II + III, I get 170 applications... this data does not match the figure.

R: Thank you for noticing this typo. We confirm that the total is 170 laser applications. We corrected this in the abstract as well.

Line 270:Chemio/hormonal...mis-typed. 

R: Thank you. Chemio was corrected to CHT and the acronym was explained in the figure caption.

Round 2

Reviewer 1 Report

Comments and Suggestions for Authors

The authors improved the paper.

However, I still have some doubts about the role of laser therapy in patients with HPV infection, especially vaginal, not to disrupt epithelial barrier and promote HPV carcinogenesis. Recent ESGO paper about management of vaginal intraepithelial neoplasia is interesting in this regard and should be cited(10.1097/LGT.0000000000000732), to some words of caution about it.

Thank you

Comments on the Quality of English Language

Minor

Author Response

The authors improved the paper.

R: Thank you for the positive feedback. We are happy to have been able to implement the constructive comments received.

However, I still have some doubts about the role of laser therapy in patients with HPV infection, especially vaginal, not to disrupt epithelial barrier and promote HPV carcinogenesis. Recent ESGO paper about management of vaginal intraepithelial neoplasia is interesting in this regard and should be cited(10.1097/LGT.0000000000000732), to some words of caution about it.

Thank you

R: Thank you for your comment, but we would like to clarify that our study focuses exclusively on the use of non-ablative laser therapy for the improvement of vaginal parameters after the treatment of the oncological lesions and not for the treatment of HPV infection or vaginal intraepithelial neoplasia.

However, we accept your suggestion to cite the recent ESGO article to provide a broader context of the use of laser therapy. We have therefore included the reference and an explanation in the manuscript: " In the gynecological field, the use of laser therapy has evolved to encompass both a curative role (for example the treatment of HPV associated lesions) as well as to improve the signs of vaginal atrophy, emerging as a potential alternative treatment for non-oncological postmenopausal VVA”. (line 54 page 2)